# A Recommendation Engine for Predicting Movie Ratings Using a Big Data Approach

**Mazhar Javed Awan** [1,*]**, Rafia Asad Khan** [2]**, Haitham Nobanee** [3,4,5,*]**, Awais Yasin** [6]**,**
**Syed Muhammad Anwar** [7]**, Usman Naseem** [8] **and Vishwa Pratap Singh** [9]

1 Department of Software Engineering, University of Management and Technology, Lahore 54770, Pakistan
2 Department of Computer Science, University of Management and Technology, Lahore 54770, Pakistan; f2019279034@umt.edu.pk
3 College of Business, Abu Dhabi University, Abu Dhabi 59911, United Arab Emirates
4 Oxford Centre for Islamic Studies, University of Oxford, Marston Rd, Headington, Oxford OX3 0EE, UK
5 School of Histories, Languages, and Cultures, University of Liverpool, 12 Abercromby Square, Liverpool L69 7WZ, UK
6 Department of Computer Engineering, National University of Technology, Islamabad 44000, Pakistan; awaisyasin@nutech.edu.pk
7 Department of Software Engineering, University of Engineering and Technology, Taxila 47050, Pakistan; s.anwar@uettaxila.edu.pk
8 School of Computer Science, The University of Sydney, Sydney, NSW 2006, Australia; usman.naseem@sydney.edu.au
9 University School of Information, Communication and Technology, Guru Gobind Indraprastha University, Delhi 110078, India; vishwapratap.phd@gmail.com
* Correspondence: mazhar.awan@umt.edu.pk (M.J.A.); nobanee@gmail.com or haitham.nobanee@adu.ac.ae or haitham.nobanee@oxcis.ac.uk or haitham.nobanee@liverpool.ac.uk (H.N.)

**Abstract:** In this era of big data, the amount of video content has dramatically increased with an exponential broadening of video streaming services. Hence, it has become very strenuous for end-users to search for their desired videos. Therefore, to attain an accurate and robust clustering of information, a hybrid algorithm was used to introduce a recommender engine with collaborative filtering using Apache Spark and machine learning (ML) libraries. In this study, we implemented a movie recommendation system based on a collaborative filtering approach using the alternating least squared (ALS) model to predict the best-rated movies. Our proposed system uses the last search data of a user regarding movie category and references this to instruct the recommender engine, thereby making a list of predictions for top ratings. The proposed study used a model-based approach of matrix factorization, the ALS algorithm along with a collaborative filtering technique, which solved the cold start, sparse, and scalability problems. In particular, we performed experimental analysis and successfully obtained minimum root mean squared errors (oRMSEs) of 0.8959 to 0.97613, approximately. Moreover, our proposed movie recommendation system showed an accuracy of 97% and predicted the top 1000 ratings for movies.

**Keywords:** recommendation engine; Spark machine learning; filtering; collaborative filtering; RMSE; Pyspark; matrix factorization; oRMSE; ALS (alternating least squared); Apache Spark; Spark ML Movielens dataset; Spark MLlib

## 1. Introduction

The Internet is becoming increasingly affordable, and therefore, Internet users are increasing daily. As a consequence, the amount of data transferred over the Internet is also increasing. This has led to a data overload, where users are flooded with knowledge and information [1]. The rapid growth in data has led to a new era of information. These data are used to build innovative, more efficient, and effective systems. In this era of big data, the recommender engine is a category among data sieving systems, which aims to

predict the ratings a user will assign to objects of interest over the Internet. Herein, a movie recommender engine is considered as an adaptation of the big data approach. Generally, a recommendation engine is a filtering-based system, which filters the search results for quality and hence provide items that are more relevant to the search items based on a user's search history [2]. These recommendation engines are used to predict the preferences of a user or the rating that a user would give to an item of interest. Such recommender systems are present in various websites and are used for information retrieval and information extraction from distributed and heterogeneous data sources [3,4].

Recently, recommendation systems have become a crucial aspect of several research areas and are used in various information retrieval platforms, such as Amazon, Netflix, and YouTube. They generate recommendations based on the user's search history and the platform, such as LinkedIn, Facebook, Amazon, and Netflix. These platforms aim to serve users by providing them with an improved experience [5]. Hence, a recommender engine is a type of data analyzing and sieving system. Such systems are becoming increasingly well known and are found in various areas, such as theatre, movies, news, books, cooking, sports, etc. Currently, most applications (such as Netflix, Spotify, and different multimedia social networks) provide engaging facilities to improve the user's experience. These applications highly depend on the effectiveness of their recommendation systems. Additionally, these recommender systems provide the users with unlimited access to the complete digital world through behavior, experiences, preferences, and interests. These systems help to analyze click-through rates and revenue for companies; they also have positive effects on the user experience and create user satisfaction. This functionality provides an improved user experience and achieves lower cancellation rates by saving costs [6].

In the movie or any recommender system (RS), the main approach is stepwise, where at first, users rate different items, and then the system makes predictions about the user's ratings for an item that has not yet been rated [7]. Thus, the new predictions are built upon the existing ratings with similar ratings of the most active user. Towards this, collaborative filtering and content-based algorithms are used most frequently. The collaborative filtering (CF) scheme enables user-based applications to reproduce personalized recommendations more quickly and effectively. The recommendation system uses machine learning algorithms to create collaborative filtering based on training data and then uses the model to make more efficient predictions [8].

Any recommendation system works more efficiently with big data, which is the driving force behind it. As data grow exponentially, it is a significant struggle to extract data from heterogeneous sources [9]. The big data in movie recommendations supply a large amount of user data, such as the behavior of all viewers, activity on the site, the ratings and filtering movies based on ratings and other viewers' data [10].

Apache Spark, which is a high-performance, openly accessible, open-source, and clustering-based computing system [11], is freely available framework and was used to analyze the big data analytics that resolves constant, iterative, or continual algorithms in the memory space. This supports an extensive set of tools developed with APIs and MLlib [12]. Spark supports a broader span of components, scope, and performance than the function used in Hadoop, i.e., MapReduce, in many supervised learning-based models [13]. Its functions use a resilient distributed dataset (RDD) for programming purposes. However, the potential strength is the execution of parallel programs. Spark provides various libraries and tools, such as Scala, Python, and R, which help to manage data and increase its scalability [14].

Our recommendation engine was built on matrix factorization and the alternating least squared (ALS) model by using Spark machine learning (ML) libraries. These machine learning application programming interfaces (APIs), which are available in Spark, are commonly known as MLlib [15]. This single library of machine learning supports multiple algorithms such as clustering, classification, frequent pattern matching, Spark ML, linear regression, linear algebra, and recommendations.

Herein, we used collaborative filtering to avoid the cold-start problem. This technique fills the missing values in the user's item-based matrix through Spark MLlib. Our proposed RS divided the rating dataset into matrices, which were item and user based along with latent features. Furthermore, to avoid the creation of a new model at every instance, we utilized elements that were extracted from the new user matrix. Here, we simply multiplied the matrix to attain the required points in less time. Through this categorization process of the recommendation engine into several collective services, the system's complexity can be minimized, and a robust collaborative system can be implemented by training continuous models. Toward this, the acquired data were given explicit ratings in the cases where the data were slightly too small. This model-based approach has one major advantage: it was able to recommend a large number of data items to a large number of users as compared to other models. It works well with sparse matrices instead of memory-based methods.

The novelty of this research is selecting several parameters of ALS that can help to build a robust recommendation system, which solves the scalability problem. The aim is to direct users to items, which are most relevant to them. This could be based on past behavior on that platform or on users who have similar profiles to a given user; therefore, the system would recommend something, which their friends like, which is the field of information extraction. In this regard, we propose a movie recommendation system using the big data approach. The main contributions of this paper are as follows:

- We propose a recommender engine that predicts top ratings for movies. Our system is implemented using collaborative filtering through Apache Spark ML libraries in cloud-based platform databricks.
- For this, we applied a model-based approach along with matrix factorization and the ALS method, which is used to more efficiently learn all latent features. It analyzed the ALS algorithm's parameters, which include rank, lambda, and iterations.
- We achieved an RMSE of 0.9761 with slightly shorter predictions than the existing methods, since they only represented the top 25 predictions.
- We further improved the root mean square error (RMSE) score, obtaining more predictions (up to 1000) with 97% accuracy, and our algorithm took just 517 s to predict these ratings with significant accuracy.

This paper is organized into six sections: Section 2 represents the literature review, background, and the context of why the system is essential in modern applications. Section 3 provides a description of the basic structure and approaches of the recommendation engine. Section 4 describes the datasets. Section 5 describes the methodology used to predict ratings and the implementation of models. Section 6 analyzes the experimental procedure and results of the implemented models. Section 7 describes the conclusion and presents some future directions.

## 2. Related Work

As the recommender system is often duped, a hybrid algorithm system was introduced by using Apache Spark to improve the efficiency of the recommender system for parallel processing. It used both structured and unstructured data by avoiding excessive information through hybrid and user-based collaboration. Significant problems were adaptability, frozen start, meagerness, and perturbability (unaccountability); the algorithm on big data can reduce the problem of scalability. However, the strength was using different combinations of collaborative filtering algorithms to overcome the freeze start problem and provide a potential recommender system [16].

The approach used was user–user, item–item, or user–item similarity, and this idea of linear combination was well implemented. Future directions for recommender systems is evolving rapidly, so in the future, the implementation of Apache Spark and R-language will be examined. The collaboration of Spark and R formats at a distributed level will make a Spark that will provide newly gathered data [17]. In another paper, scalability was discussed and resolved by using the query-based approach with the help of Apache Hivemall, which evaluates similarities and dissimilarities on each node. Consequently,

non-parallelism within this issue was common, but using the collaborative approach, the speed and the growth of datasets, the scalability issue was resolved and had improved extensibility and execution time than the sequential method. The extensibility of the system, the speedup, and execution-time of each node was recorded, and the results showed that the total execution time of the multi-criteria methodology using Hive was far superior to the single consequent non-parallel approach. The knowledge gap was the one constraint that needs to be resolved in the communication process within the network for higher efficiency [18].

In this paper, the researcher described Spark terminology used for real-time execution with combined algorithms, based on two modes—user-level and item-level. Two more algorithmic techniques were used to resolve the constraint of deficiency, sparsity, and extensibility [19]. Recommender systems helped to overcome the issue of excessive flooding of information based on the history of a user's likes and dislikes by providing suggestions. The literature review of this paper concerned various methods of how features can be extracted. The study discussed ALS (altering least square), Scala programming language, and Clustering K-means; the last two were the main algorithms used to control the limitations of CF extensibility sparseness [20]. The study ML libraries were used to build a required recommendation system, but the lambda system also seemed capable of handling such a large amount of data. Lambda structures, SparkML libs, direct and indirect data collection, auto analysis techniques, and different training models were used to perform a real-time analysis of big data [21].

Recently, machine learning (ML)-based models [22–24] have been implemented in various domains, which discussed time limitations, specification, accuracy with efficacy, and intricate issues. However, the potential strengths were auto analysis, high adaptability, and user data collection from big data environments through recommendation. The experimental results showed that Apache Spark and machine learning collect data about viewers' preferences and processes; the results provide highly accurate recommendations in listed form. The recommender system helps users to select items from the huge number of options on social service applications, such as Netflix and YouTube, for user ease [25].

In 2021, the authors implemented stock market [26], Black Friday sale [27] and Amazon food review [28] predictions using Naive Bayes, and other machine learning models on Spark MLlib achieved above 90% accuracy with a good running time. The author developed a system that described how meaningful information could be extracted from gigantic raw data, with the implementation of several algorithms based on data mining. The experimental results showed authentic results with larger datasets in all circumstances if accurate models were obtained [29]. The study compared four different RS models based on a combination of user similarity and trust weighted propagation. The accuracy of back propagation and singular value decomposition was under 70%. The problem was the high value of the optimal score recommendation only [30].

The most common framework that was implemented in this paper is Spark and then ALS (altering least squares), Lambda architecture, and the K-means algorithm, which was utilized mainly for clustering purposes. The researcher noticed that the weakness of this was the unnecessary sequential comparison of the smaller dataset, although the strength was the Spark machine learning libraries. The experimental results, which were taken from comparing the MSE (mean squared error) and WCSS (sum of cluster squared error), showed that the real-time processing of the size of datasets could be improved by enhancing the WCSS by growing various clusters. Evaluating two algorithms could provide improved results as the number of clusters was increasing and needed further optimization [31].

Generally, they discussed two types of Spark ML-based strainer systems. The first was specificity based, and the second was interactive or hybrid based. The future directions were given by discussing more advanced algorithms, such as matrix factorization, which would enhance the specificity and accuracy, and rapidity based on previous content used by the user and in-memory utilization [32]. In this study, they examined and evaluated

models and the Spark-based SARF methodology using a huge amount of data by matching queries to users with correct results and higher quality. With the recent growth in data, the search recommendation engine is used to resolve excessive information issues. In this paper, the SARF algorithm was used, which built joins, resilient distributed datasets (RDD), and filters based on Spark. The main issues are cold start, deficiency, and sparseness problems; CF was used with a hybrid algorithm that provides accurate data, and it is likely that it also resolved the problem of cold start [33].

As data are increasing day-by-day, to analyze these huge volumes of datasets, and for the convenience of the user, a small enhanced application of the suggestion system is the integration of the analytics of big data whose base was built upon the Spark framework with several combined CF algorithms. The frameworks used in this paper, for data streaming purposes and machine learning libraries, with graphs, SQL, matrices, Spark's join, filters, RDDs, and MapReduce functions are discussed in this paper. Weaknesses were scantiness and strength within the improved version of ALS that helped to obtain more accurate results [34]. This paper introduced a significant approach in this big data era, attained the desired suggestions by the recommender system, and presented the ALS model with several Spark machine learning libraries (Spark-ML and MLlib). Related work or an algorithm/approach CF was used but for filtering purposes only. In this paper, the ALS model was used for predicting movies. This paper introduced a significant approach in this big data era to attain the most-desired suggestions by recommender systems; these ALS models with several Spark machine learning libraries (Spark-ML) were introduced. The model in this study outperformed the existing model and improved the root mean squared error (RMSE) by approximately 0.97613. Additionally, it showed an accuracy of 97%, outperforming the state-of-the-art, which had only 91%. Moreover, the methodology, performance evaluation, and performance analysis were conducted on the dataset taken from Movielens. It concluded that analytics based on collaborative filtering using the ALS model reduced the common sparsity problem [35].

Our proposed RS is based on user-level and item-level filtering, which is derived from collaborative filtering through the big data framework Spark. Although the dataset is large, we used ALS that categorized the rating dataset into a matrix that was item-based and user-based along with latent features. Thus, when we recombined them, we obtained a new matrix by multiplying the divided matrices. The new matrix may have a missing entry that represents the movies and predictions and may cause a cold-start problem. If there are any missing entries, the system fills the space based on user and item-level similarities. To avoid the creation of a new model every time, we utilized elements that are extracted from the new user matrix that simply multiplied the matrix to obtain the points in a much quicker time. Experimental results show that the outcomes of the proposed algorithm are productive and successful. Additionally, it provides quick, top-rated predictions of movies. In future investigations, we need to focus on more mining-related approaches to obtain knowledge about user-related paradigms.

## 3. Basics of a Recommender System

A recommender system is an automated system with which to filter some entities. These entities can be any products, ads, people, books, songs, or movies. These are derived from all platforms daily, from Amazon to Netflix, Pandora, YouTube, and eHarmony. There are three approaches commonly used by recommender engines; one is content-based filtering, the other is collaborative filtering, and an approach that some of the recommender systems also use is demographic filtering, a mixture of both of these approaches. This study developed an RS based on collaborative filtering by building and evaluating various models that predict top-rated movies for users. Figure 1 shows the recommender system.

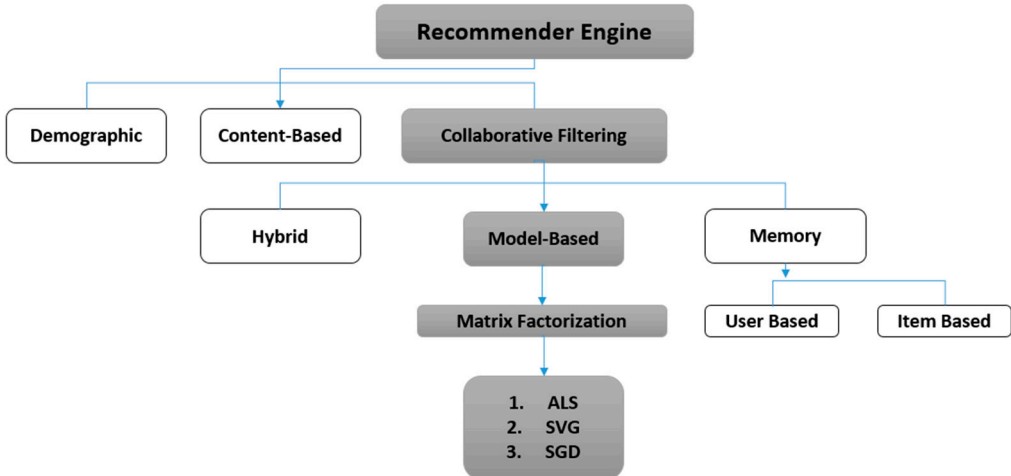

**Figure 1.** Basic structure of the recommendation engine.

### 3.1. Collaborative Filtering

This is a technique which is used by recommendation systems. This filters out elements that the user might have reacted to. Based on similarities, it finds different large groups of people and a smaller number of users; then, the items that they like create a list of suggestions [36].

It integrates the same users and their choices to build a list of suggestions. The suggestions that are given by this filtering technique are built on the automatic cooperation of several users and their close interests. All the users' choices are compared, and they receive a recommendation. For example, when you purchase something online, watch a movie, or even read an article, you are often given a chance to rate that item on a scale of one to five stars, a thumbs up, or some other type of rating system; based on your feedback from these types of rating systems, companies can gather a large amount of information about your feedback and your preferences and make predictions about your preferences and offer recommendations based on ratings from users that are similar to you. Figure 2 shows a basic diagram of the collaborative filtering method. Suppose a recommendation engine that recommends videos, such as standard services by YouTube and Netflix, is used for this purpose.

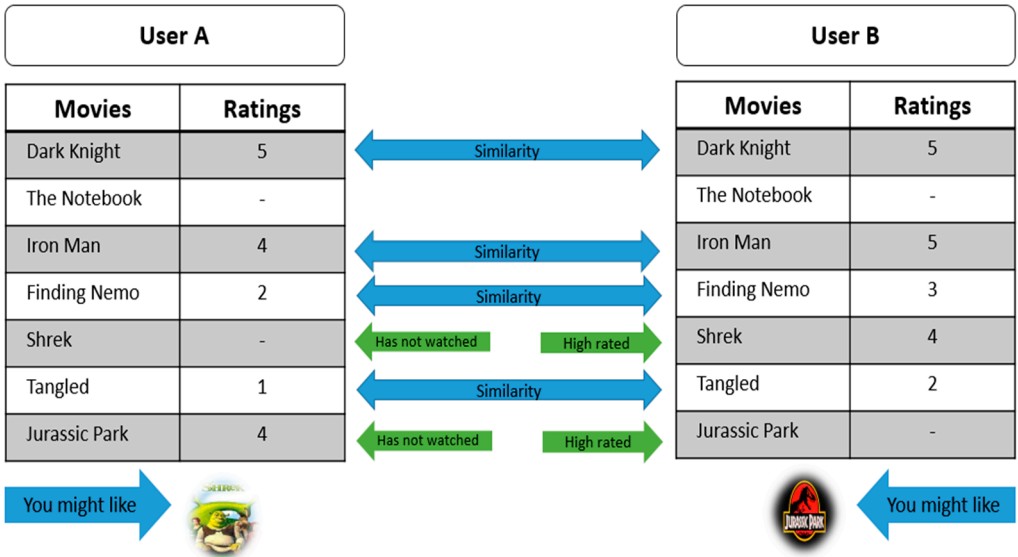

**Figure 2.** Basic structure of collaborative filtering.

For this, users first subscribe to their services and channels. Then, users that have similar preferences will be grouped, and the new most popular video is then recommended to that particular group of users. Figure 2 shows that multiple users use collaborative filtering, and each entry in this figure indicates how many movies have been watched by how many users. If User A watched *The Dark Knight* and *Iron Man* and gave a high ranking but did not like *Tangled* and *Finding Nemo*, User B, who also liked *The Dark Knight* and *Iron Man*, would not like *Finding Nemo* and *Tangled* as much because they have similar preferences; if one of them gave a high rating to a movie, then we could infer that the other user that has not yet it seen is likely to enjoy that movie as well. Based on this logic, we can offer a recommendation [37].

### 3.2. Cognitive Filtering (Content-Based)

This approach is also known as a cognitive technique, based on a previous search, explicit feedback (or the history of the user), the user profile, and what the user likes and reacts to; all this information is taken into account for the suggestions or recommendations. Figure 3 describes that if the user[1] or can say any user watches a video in the comedy category, then it is most likely that a video under the comedy genre would be recommended to the user in the future. The recommendation engine independently recommends videos completely based on the user's behavior, but not by the behavior of other users [38,39].

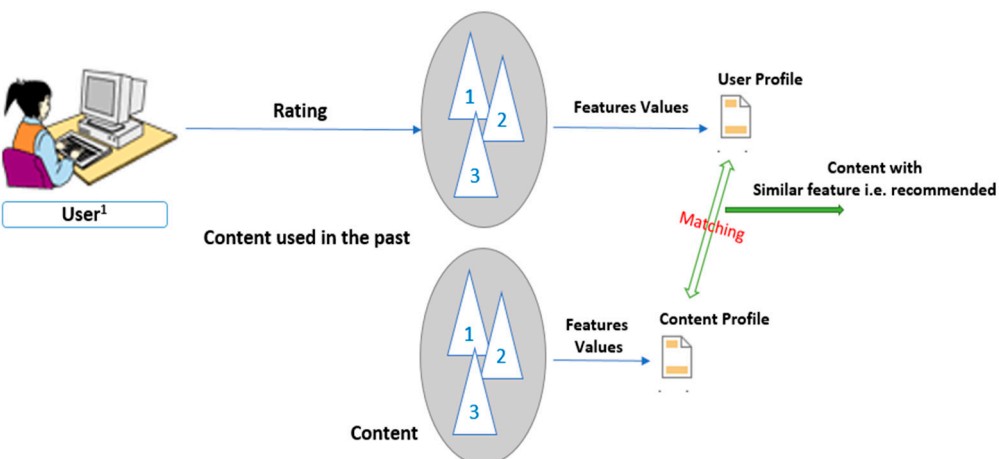

**Figure 3.** Content-based filtering mechanism.

### 3.3. Demographic Filtering

Demographic filtering is an approach that offers hypothesized recommendations based on a popular and/or group class or category for each user. The recommender system suggests similar series/movies to the users along with the same demographic characteristics. Though each user is distinct from others, this technique is observed to be very straightforward and easy. Figure 4 shows demographic filtering, which describes the main procedure followed by the system that registers the most popular and unpopular series that have a high possibility of being liked, so that there is a possibility of them being enjoyed by a large number observers based on any user [40].

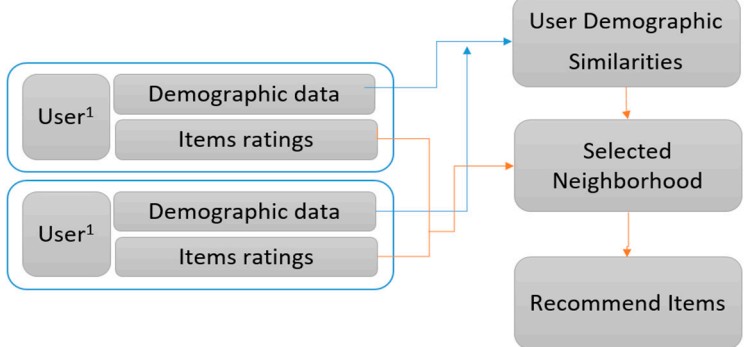

**Figure 4.** Demographic-based approach of any user.

### 4. Datasets

This platform uses a 100 k lens to test and analyze the recommender engine. Movie recommendation sites use rating and reference systems used globally to categorize a movie based on its content. Therefore, these systems are useless in social networking recommendation systems. Different sites use the user's preferences in content and movies on TV, theatre, or VCR. Other sites use different approaches, such as creating social networking rating facilities and recommending movies that users have liked or referred to. We used Movielens, a classical, user rating and community user dataset, for our experiments [41].

### 5. Methodology

The technique used in this paper is collaborative filtering, which is based on ALS, i.e., a matrix factorization algorithm that is used to predict the top-rated movies that will be recommended to the users later. This takes a large matrix and factorizes it into a small depiction matrix with the help of ALS. The description of the dataset and all experimental evaluations is shown below in Section 5.

#### 5.1. Collaborative Filtering

Typically, collaborative filtering is used for recommender engines. The technique uses matrix factorization to predict the missing entries. In this study, the rating for the movies implemented using Spark-ML and ML-libraries, which currently use the model-based collaborative scheme, can predict missing entries, in which users or products are modeled by latent factors. Moreover, we implemented the ALS (alternating least squared) algorithm to learn these latent factors.

#### 5.2. Matrix Factorization

Matrix factorization is used to resolve the sparsity problem in collaborative filtering. It is one of the models of matrix factorization that is considered as the values from the user's item list. Selecting several parameters to perform analytics on the ALS algorithm helps in building an efficient final recommendation system. This paper implemented the recommendation system using the ALS algorithm with Apache Spark on databricks. Figure 5 depicts the proposed RS which is based on the user-level and item-level techniques, utilized with a matrix factorization approach along with ALS that divided the rating dataset into a matrix that was item-based and user-based along with Latent features. Thus, when we recombined them, we got a new user matrix. The new matrix may have a missing entry. If there is any missing entry, the system will fill the space based on user and item level similarities. It will re-fill the values based on U-matrix.

The performance evaluation can be measured using root mean squared error (RMSE) for rating the prediction and the model will be trained.

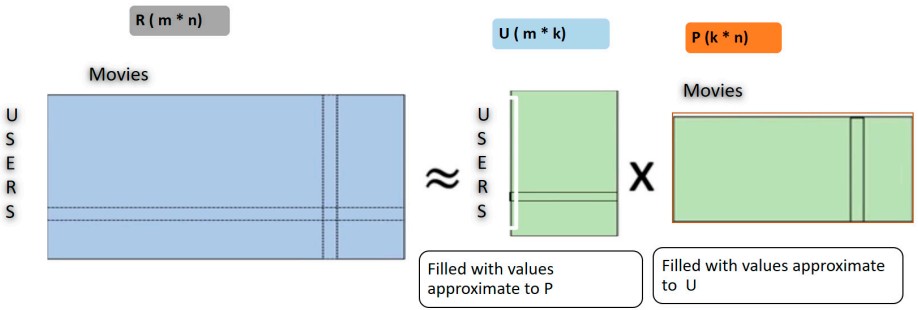

**Figure 5.** Working diagram of matrix-factorization method used in ALS.

*5.3. Architecture of ALS (i.e., Regression Evaluator)*

This study is based on machine learning along with the ALS algorithm, which is a similar kind of regression analysis where regression presents a line of data points that are used to draw through regression. In this way, the sum of squares of the distance from these displayed data points reduces the distance. These lines are used to predict the function's value, which is then met with the independent variables [42].

In a matrix of users, in which the users provide ratings for the movies they have watched, most of the matrix is empty, as there are far too many movies for anybody to reasonably watch; one of the major benefits of ALS is that it works well with such a sparse matrix, but it seeks to leverage ratings from a similar user. The users fill all the blank cells with predictions of how these users would rate these movies if they were to watch them; the predictions with the highest values then become recommendations that can be offered to users, as shown in Figure 6.

| | Movie 1 | Movie 2 | Movie .. | Movie N | | | Movie 1 | Movie 2 | Movie .. | Movie N |
|---|---|---|---|---|---|---|---|---|---|---|
| User 1 | 1 | Blank | Blank | 3 | | User 1 | 1 | 4 | 2 | 3 |
| User 2 | Blank | 5 | Blank | 3 | | User 2 | 1 | 5 | 3 | 3 |
| User 3 | Blank | Blank | 1 | Blank | | User 3 | 2.5 | 2.8 | 1 | 3.5 |
| User 4 | 2 | 3 | Blank | Blank | ALS | User 4 | 2 | 3 | 2 | 3.5 |
| User 5 | Blank | Blank | 1 | Blank | | User 5 | 2.5 | 2.8 | 1 | 3.1 |
| User .. | Blank | 3 | Blank | Blank | | User .. | 2 | 3 | 2 | 3 |
| User M | Blank | Blank | Blank | 4 | | User M | 1 | 4 | 2 | 4 |

**Figure 6.** The process of the ALS algorithm.

Suppose we have a Matrix R, ALS will take that matrix and factor it into two smaller matrices, U and P, where x together produces an approximation Matrix R. The way the ALS decides what values to put into U and P is by first filling in the random numbers and calculating the error term according to this error formula; then, alternating back and forth between Matrix U and Matrix P, the ALS adjust the matrices to iteratively decrease the error, and it continues to alternate between them until the error term is minimized. Once this is completed, the Matrices U and P are multiplied back together. The benefit of ALS is that, when the matrices are multiplied back together, the blank spaces in the original Matrix R are filled [43].

The general structure for the RS follows various phases, as shown in Figure 7. It shows the process of the proposed system, which is a recommendation system employed over the collaborative filtering approach combined with a model derived from a matrix factorization scheme. It depicted that there is a huge amount of data that can be taken as input (e.g., streaming data, user rating data, movie data, and community-related data).

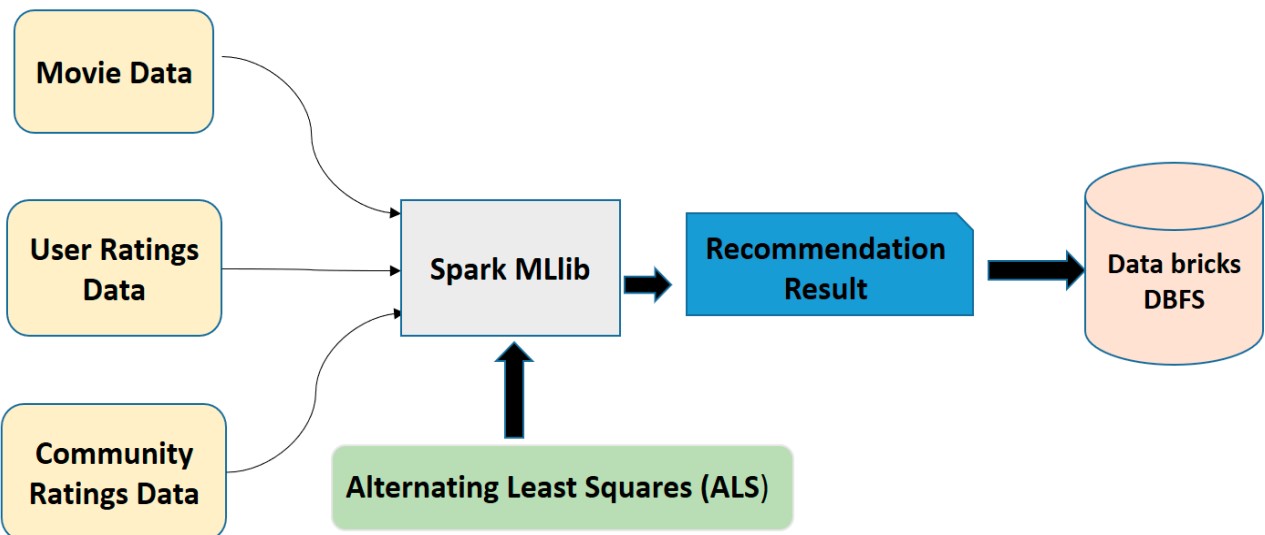

**Figure 7.** Block diagram of the proposed system using ALS through Apache Spark-MLlib.

First, the system selected a large dataset derived from users' behavior and took it as input. We applied Apache Spark MLlib using Python in the databricks platform, which is free and user-friendly. The next step was to train the dataset, pre-process the data, and evaluate the model (i.e., ALS). We applied a model called alternating least squares based on matrix factorization; it parsed the model through a regression evaluator. Then, it trained the dataset, completed data processing, evaluated the algorithm, and produced recommendations by fetching the results. Thus, we obtained two matrices that were equal to the original one but with small dimensions. After evaluating the model, it produced the recommendations and predictions of ratings. Then, the output was fetched from the databricks database through Spark MLlib. The results of the experimental analysis are discussed in the section below. Section 6 describes the results achieved by the proposed system.

## 6. Experimental Results and Discussion

In this paper, we examined the Movielens dataset and evaluated results by implementing the ALS algorithm, and to evaluate all the experimental analysis and results, the environment used is databricks a cloud-based platform, and Google Colab [44] used for scaling a massive amount of data which created oCs (optimized clusters) for Apache Spark for the prediction of ratings for the user.

This study aimed to find the best top-rated movie predictions and improve the root mean squared error (RMSE). The effectiveness of the recommendation was evaluated using the root mean squared error (RMSE); the idea is that the smaller the RMSE of the recommender system, the more efficient the recommendation. This section is divided into two parts: the first part describes the dataset, while the second part represents all the experiments and results after applying the model.

### 6.1. Dataset Description

The evaluation and performance analysis of the proposed technique was performed on the dataset taken from Movielens. It consists of two files; the first one is movies.dat, which contains 10,681 movies, and the second file is rating.dat, which contains 1 million rating parameters from 5 k users on 10,681 movies (data description shown in Table 1).

**Table 1.** A brief description of the Movielens dataset.

| Summary | User ID | Movie ID | Rating | Timestamp |
|---|---|---|---|---|
| Count | 299,771 | 299,771 | 299,771 | 299,771 |
| Mean | 3021.053196866942 | 1867.84192266753634 | 3.5793755900337256 | $9.722794929589853 \times 10^8$ |
| Std. dev. | 1726.5019734897063 | 1095.567508110294 | 1.1192410652643803 | $1.218013356846537 \times 10^7$ |
| Min | 1 | 1 | 1 | $9.56703954 \times 10^8$ |
| Max | 5041 | 3953 | 5 | $1.04645459 \times 10^9$ |

*6.2. System Implementation*

The following steps were used to implement our proposed system:

- First, we created clusters in the experimental environment called "databricks": a cloud-based platform which launches several optimized clusters for the Spark using the language Python.
- We used CF along with ALS and predicted the rating for users specifically based on users' ratings.
- Then, the obtained predictions were collaborated with the other ratings of the users.
- We fetched all the results from databricks through Spark ML.
- We performed two experiments with different divisions of datasets.
- From the first experiment, the system produced predictions with an RMSE of 0.9761 by outperforming the existing one which had 0.91.
- Additionally, from the second experiment, we achieved an oRMSE of approximately 0.8959, which is the minimum in this case, and predicted ratings of movies up to 1000.

Experiment 1

The original datasets for rating the movies split the RDD into three categories—training, testing, and validation sets—in the ratio of 60:20:20, i.e., 60% for the training models, 20% for the validation of the models, and the remaining 20% for testing. In this model, the root mean square (RMS) is used to evaluate the performance factor. It calculates the error rate that a user assigns to the system and predicts the error of the model.

$$\text{RMSE} = \sqrt{\sum_{i=0}^{n} \frac{(A_{xi} - B_{yi})^2}{n}} \qquad (1)$$

In the above equation, $Ax_i$ is the rating that user $x_i$ assigns to the item, along with $B_{yi}$, which is the rating after prediction that the user assigns to the item, and lastly, $n$ is the number of ratings in the test data calculated using the regression evaluator, and defining a function that evaluates the RMSE. Based on the validation RMSE, we defined a function to perform a grid search and find the ALS model, then applying ALS to find movie rating predictions. Tables 2–4 show the results of movie lists vs. genres, movie prediction against the movie ID, and top five movie ratings, respectively. When using the mean rating of each movie as the prediction, the testing RMSE[1] is 0.9761; Figure 8 shows the results for the training and test sets.

**Table 2.** Movie lists according to different genres.

| Movie ID | Title | Genres |
|---|---|---|
| 1 | *Toy Story (1995)* | Adventure \| Animation |
| 2 | *Jumanji (1995)* | Adventure \| Children |
| 3 | *Grumpier Old men* | Comedy \| Romance |
| 4 | *Waiting to Exhale* | Comedy \| Drama \| Romance |
| 5 | *Father of the Bride* | Comedy |

**Table 3.** Movies predictions for Movie IDs.

| Movie ID | Prediction |
| --- | --- |
| 1580 | 3.5247 |
| 2366 | 3.5937 |
| 3175 | 3.6931 |
| 1088 | 3.2884 |
| 32,460 | 3.7500 |

**Table 4.** This table only shows the top five movie rating predictions.

| User ID | Movie ID | Rating | Prediction |
| --- | --- | --- | --- |
| 555 | 1580 | 2.0 | 3.5247 |
| 520 | 1580 | 5.0 | 3.52475 |
| 450 | 1580 | 3.0 | 3.5247 |
| 415 | 1580 | 3.5 | 3.5247 |
| 369 | 1580 | 1.0 | 3.5247 |

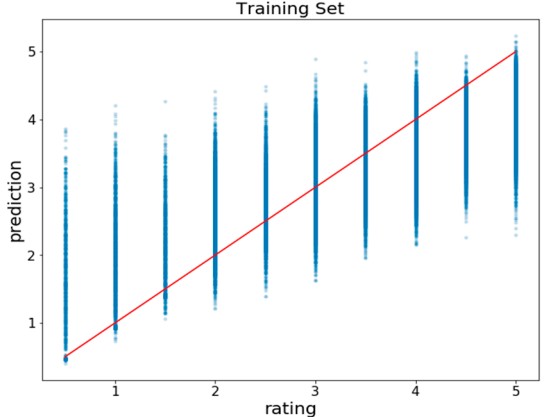 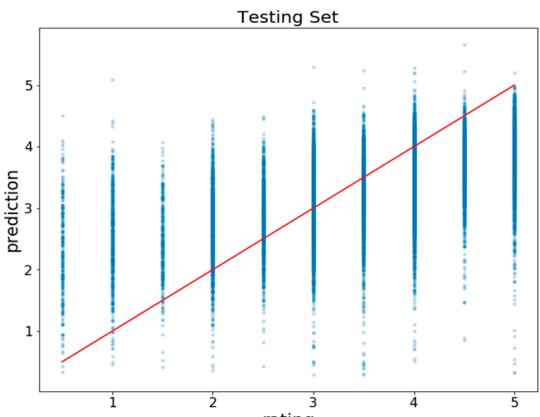

**Figure 8.** Graphical representation of training and testing set predictions.

### 6.3. Experiment 2: oRMSE Analysis and Performance Evaluation

Furthermore, performance evaluation and analysis were conducted for the dataset. This section presents a detailed description of the analysis of the dataset and is organized into several sub-sections. First, we loaded and verified the dataset, as shown in its description. We then divided it into two different sizes of training and testing categories with a ratio of 7:3.

The original datasets for rating the movies split the RDD into two elements, training and testing, in a ratio of 0.7:0.3, i.e., 70% for the training models, and 30% for the testing. In the second experiment, we examined the *RMS* score of the previous model by making some modifications and evaluated the performance factor. This obtains the error rate that a user assigns to the system and predicts the error, as shown in Figure 9. We applied the alternating algorithm by training the dataset and evaluated the model.

Experimental Evaluation

In our experiments, the best model has six latent factors and regularization = 0.2 with iteration = 5. The results from the experimental evaluation are represented in Table 5. It shows iterations with various latent factor values (06, 08, 10, and 12, respectively). It also describes regularization values, 0.5, 0.1, 0.2, 0.4, and 0.4, and validates the root mean square estimation (VRMSE) score regarding each factor value.

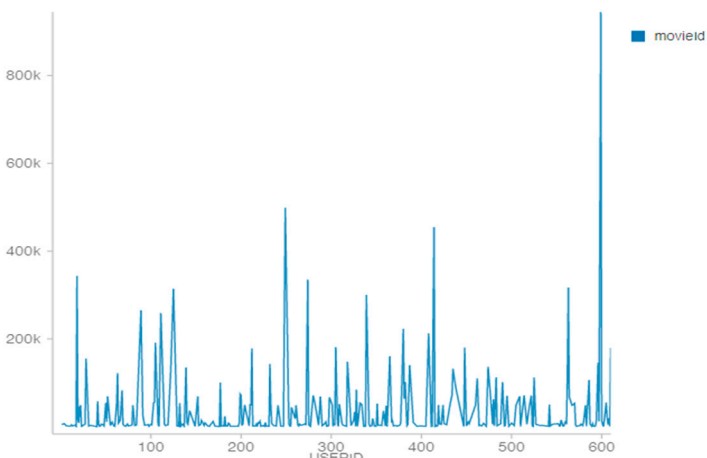

**Figure 9.** The area covered by the results after predicting the ratings.

**Table 5.** The number of iterations with regularization with different RMSE parameter values.

| Iter: 05 AND LF: 06 | | Iter: 05 and LF: 08 | | Iter: 05 and LF: 10 | | Iter: 05 and LF: 12 | |
|---|---|---|---|---|---|---|---|
| **Reg.** | **VRMSE** | **Reg.** | **VRMSE** | **Reg.** | **VRMSE** | **Reg.** | **VRMSE** |
| 0.05 | 0.9774 | 0.05 | 0.9911 | 0.05 | 0.9978 | 0.05 | 1.0053 |
| 0.1 | 0.9129 | 0.1 | 0.9168 | 0.1 | 0.91766 | 0.1 | 0.9177 |
| 0.2 | 0.89515 | 0.2 | 0.8984 | 0.2 | 0.8987 | 0.2 | 0.9000 |
| 0.4 | 0.9694 | 0.4 | 0.9702 | 0.4 | 0.9695 | 0.4 | 0.9701 |
| 0.8 | 1.1934 | 0.8 | 1.1936 | 0.8 | 1.1935 | 0.8 | 1.1934 |

The best model has six latent factors and regularization = 0.2: training RMSE is 0.6876; validation RMSE is 0.8951 with an optimized runtime of 517.30 s. We further obtained a testing RMSE of 0.8959. We utilized the model and monitored the precision, training, and validation testing scores, which predicted the ratings of users and obtained the best results for the different numbers of parameters and iterations. Thus, through experimental analysis, we examined and evaluated all the results; it was concluded that the selection of parameters related to alternating least squares could affect the performance of the RS that will be used. This study shows that Apache Spark MLlib along with big data analytics produced more robust and accurate results. We obtained 97% accuracy in our proposed recommender engine (RE) than the existing one, which had an RMSE of approximately 0.91. There is a huge difference between the results. Additionally, the employed model achieved the top ratings of the movies and produced results of the predictions up to 1000 movies, but only the top 35 movie predictions can be seen in Table 6.

In this work, the proposed robust recommender system predicted user ratings based on the user's behavior. The resultant prediction (Table 7) shows the top 35 movies, although we achieved the top 1000 best predictions. It performed well in each iteration, because it took very little time to provide the above results for up to 1000 rows. Table 7 shows the results obtained in both experiments in terms of accuracy, oRMSE$_1$, oRMSE$_2$, and prediction.

Thus, in the first experiment, we obtained an RMSE of 0.9761 but with slightly fewer predictions than the existing one—25 predictions. Additionally, we improved the oRMSE score, and to obtain more predictions, we performed another experiment, which achieved improved results and predicted ratings up to 1000, with a 97% accuracy but with a slightly lower RMSE score; however, this did not affect the performance of the system because it only took 517 s to predict the ratings with remarkable accuracy,. From the experimental analysis, we observed that including a recommendation in the current system will be efficient and attractive because it enables the experience to involve more user and customer engagement. Additionally, we performed two experiments; in each, we obtained

predictions with improved accuracy, and the RMSE score can be seen in Table 7, compared to the existing study, which had a minimum RMSE of 0.91, but the accuracy of each iteration or system is not shown. Our proposed system obtained 97% accuracy with the top 35–1000 predictions with an optimized RMSE score of 0.89–0.97. We suggest that the machine learning algorithm works vigorously in these systems for large datasets, and these algorithms can be used to predict the values based on demographic content. We calculated explicit and implicit ratings inferring the preferences of the user and presented a graphical representation of the results, as shown in Figure 10.

**Table 6.** The top 35 predictions achieved for up to 1000 movies.

|  | User ID | Movie ID | Rating | Prediction |
|---|---|---|---|---|
| 1 | 122 | 858 | 5 | 4.9316 |
| 2 | 171 | 858 | 5 | 4.9231 |
| 3 | 122 | 7387 | 5 | 4.8304 |
| 4 | 12 | 2572 | 5 | 4.8156 |
| 5 | 122 | 48780 | 5 | 4.8061 |
| 6 | 43 | 1084 | 5 | 4.7963 |
| 7 | 603 | 1699 | 5 | 4.7840 |
| 8 | 452 | 1270 | 4 | 4.7304 |
| 9 | 122 | 69481 | 4.5 | 4.7247 |
| 10 | 1 | 1270 | 5 | 4.6855 |
| 11 | 51 | 3918 | 5 | 4.6535 |
| 12 | 348 | 858 | 5 | 4.6352 |
| 13 | 435 | 858 | 5 | 4.6152 |
| 14 | 435 | 48780 | 5 | 4.6046 |
| 15 | 169 | 1270 | 4 | 4.5921 |
| 16 | 472 | 858 | 5 | 4.5662 |
| 17 | 543 | 1580 | 5 | 4.5361 |
| 18 | 236 | 2580 | 5 | 4.5354 |
| 19 | 339 | 17675 | 5 | 4.5274 |
| 20 | 417 | 858 | 5 | 4.5272 |
| 21 | 80 | 48780 | 5 | 4.4892 |
| 22 | 200 | 54190 | 5 | 4.4816 |
| 23 | 52 | 69481 | 4 | 4.4683 |
| 24 | 74 | 858 | 5 | 4.4402 |
| 25 | 587 | 1270 | 5 | 4.4319 |
| 26 | 532 | 858 | 5 | 4.4201 |
| 27 | 336 | 958 | 4.5 | 4.4126 |
| 28 | 572 | 858 | 5 | 4.4101 |
| 29 | 221 | 858 | 4 | 4.4047 |
| 30 | 122 | 858 | 4.5 | 4.9316 |
| 31 | 607 | 858 | 3.5 | 3.9248 |
| 32 | 474 | 6620 | 4 | 3.9240 |
| 33 | 606 | 1270 | 3.5 | 3.9210 |
| 34 | 577 | 1303 | 5 | 3.9208 |
| 35 | 582 | 4878 | 3 | 3.9171 |

**Table 7.** Comparison table of oRMSE obtained in our experiments.

|  | RMSE | Accuracy | Prediction |
|---|---|---|---|
| $oRMSE_1$ in exp1 | 0.9761 | 93.5% | 5 |
| $oRMSE_2$ in exp2 | 0.8959 | 97.0% | 1000 |
| Existing RMSE | 0.9100 | 91.0% | 25 |

In the future, we will use deep learning-based approaches to apply movie recommendations. Various recent studies have been inspired by multimodal-based approaches with efficient deep learning and BigDL framework [45–52]. Moreover, the proposed system can also be applied for books or news recommendations.

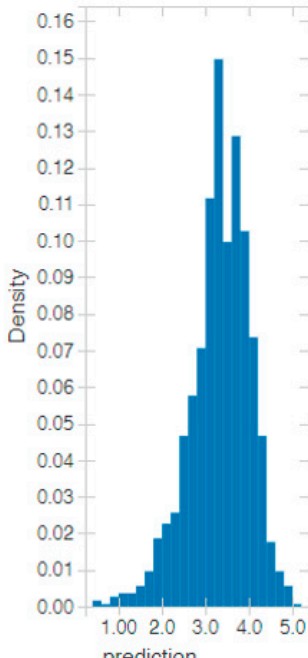
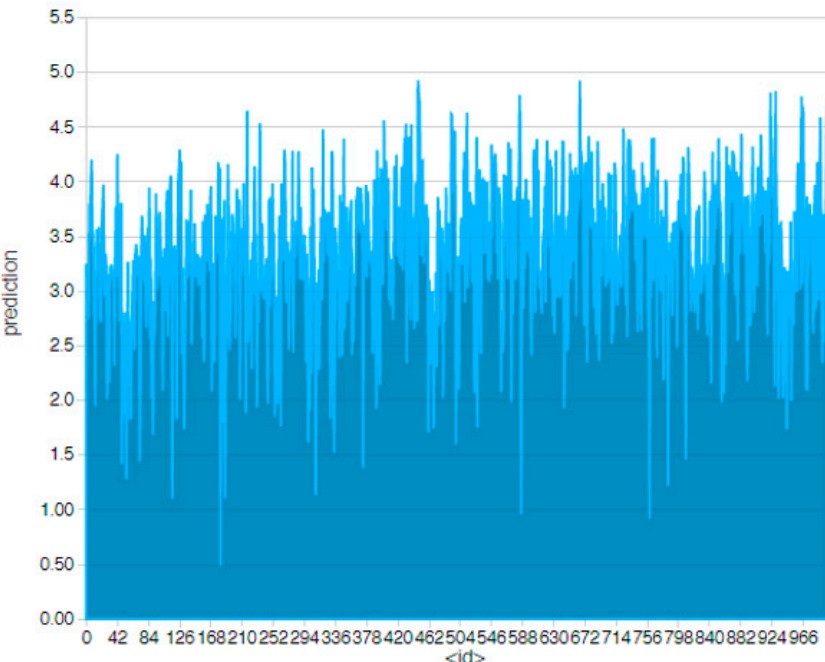

**Figure 10.** Graphical representation of the 1000 predictions.

## 7. Conclusions

We proposed a recommendation engine to predict top-rated movies based on the collaborative filtering approach through experimental analysis. The performance evaluation and performance analysis were performed on the dataset that is taken from Movielens. In this paper, we applied the ALS (alternating least squared) algorithm problem with CF, which solved the scalability, sparsity, and cold-start problem. We compared the results of our developed model with the state of the art by performing two experiments with different divisions of modules in the dataset. The first experimental results were achieved along with an RMSE of 0.97613. The second iteration obtained predictions of up to 1000 movies and a minimum root mean squared error (oRMSE) of 0.8951, and improved the accuracy by 97%. After obtaining the results, it was concluded that the selection of parameters related to alternating least squares could affect the performance of RS. Thus, a system must have a backup to memorize the user preferences for the recollection and gathering of precise and comprehensive data. In the future, sieves can be extended for highly improved results.

**Author Contributions:** All authors contributed equally to this work; conceptualization, M.J.A., R.A.K., H.N., A.Y., S.M.A., U.N., and V.P.S.; methodology, M.J.A., R.A.K., H.N., A.Y., S.M.A., U.N., and V.P.S.; software, M.J.A., R.A.K., A.Y., S.M.A., U.N., and V.P.S.; validation, M.J.A., R.A.K., H.N., A.Y., S.M.A., U.N., and V.P.S.; formal analysis, M.J.A., R.A.K., A.Y., S.M.A., U.N., and V.P.S.; investigation, M.J.A., R.A.K., H.N., A.Y., S.M.A., U.N., and V.P.S.; resources, M.J.A., R.A.K., H.N., A.Y., S.M.A., U.N., and V.P.S.; data curation, M.J.A., R.A.K., A.Y., S.M.A., U.N., and V.P.S.; writing—original draft preparation, M.J.A., R.A.K., A.Y., S.M.A., U.N., and V.P.S.; writing—review and editing, M.J.A. and H.N.; visualization, M.J.A., R.A.K., H.N., A.Y., S.M.A., U.N., and V.P.S.; supervision, H.N. and M.J.A.; project administration, H.N. and M.J.A.; funding acquisition, H.N. All authors have read and agreed to the published version of the manuscript.

**Funding:** This research received no external funding.

**Data Availability Statement:** https://grouplens.org/datasets/movielens/, accessed on 15 March 2021.

**Conflicts of Interest:** The authors declare no conflict of interest.

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
