# Peer review of "A Recommendation Engine for Predicting Movie Ratings Using a Big Data Approach"

_electronics, doi:10.3390/electronics10101215_

Round 1

Reviewer 1 Report

In this era of big data, the amount of video content has dramatically increased with the

exponential broadening of streaming services such as Netflix and YouTube. Hence, it has become very strenuous for end-users to get the desired videos. Therefore, to attain an accurate and robust

clustering of information, a hybrid algorithm is used to introduce the recommender engine with collaborative filtering using the apache spark and ML libraries. Machine learning (ML) has become an effective tool with the recent growth of diverse techniques that have been proven helpful for

various applications. Commonly there are three filtering methods: content-based collaborative filtering (CF), and demographic filtering. This study implemented a collaborative filtering method using ALS (Alternating Least Squared) model to predict the best-rated movies The paper is interesting overall, but the following are the comments that must be addressed:

 Comments:

English should be corrected.

  • Please add a block diagram of the proposed research step by step, what is the result of the paper?
  • The technique used in this paper is collaborative filtering which is based on ALS. If the data volume is very large, the transformation will become a super-slow process. Please give out the details about the transformation and address how to handle a large amount of data in this step.
  • Section 1 introduction part is a most important part of manuscript and authors should need to update the structure of that part that will be good for readers such as AI>>ML>proposed approach with artificial intelligence

References should be 2018-2021 Web of Science about 50% or more

Khadse, Vivek P., Syed Muzamil Basha, N. Iyengar, and D. R. Caytiles. "Recommendation Engine for Predicting Best Rated Movies." International Journal of Advanced Science and Technology 110 (2018): 65-76. Khan, M.A., Karim, M. and Kim, Y., 2018. A two-stage big data analytics framework with real world applications using spark machine learning and long Short-term memory network. Symmetry, 10(10), p.485.

Khalid, Z. M., & Zebaree, S. R. (2021). Big Data Analysis for Data Visualization: A Review. International Journal of Science and Business, 5(2), 64-75.

Choudhury, S. S., Mohanty, S. N., & Jagadev, A. K. (2021). Multimodal trust based recommender system with machine learning approaches for movie recommendation. International Journal of Information Technology, 1-8.

.The major contribution of the paper looks very weak so authors should think about it deeply as given Line98~101

  • How do you measure the success of a recommendation? here is the data complexity issue, the authors did not discuss complex issues in the manuscript.
  • Before Conclusion, please draw a Table and compare with previous researchers, how your approach is better in terms of accuracy.
  • Authors miss experiment setup, without experimental setup results are doubtful please explain the experimental environment in detail. few parameters are missing??
  • Conclusion: point out what are you done.

-is there a possibility to use the proposed method for other problems?

Author Response

Reviewer 1:

Comment

In this era of big data, the amount of video content has dramatically increased with the exponential broadening of streaming services such as Netflix and YouTube. Hence, it has become very strenuous for end-users to get the desired videos. Therefore, to attain an accurate and robust clustering of information, a hybrid algorithm is used to introduce the recommender engine with collaborative filtering using the apache spark and ML libraries. Machine learning (ML) has become an effective tool with the recent growth of diverse techniques that have been proven helpful for various applications. Commonly there are three filtering methods: content-based collaborative filtering (CF), and demographic filtering. This study implemented a collaborative filtering method using ALS (Alternating Least Squared) model to predict the best-rated movies.

Response: We would like to thank the reviewer for their time and the encouraging remarks. The comments have been incorporated, with details presented below. Furthermore, we improved the introduction section, language, styles, and figures to make the paper useful for readers. Once again, we do appreciate all your positive comments and valuable suggestions; and believe that these helped us in improving the quality of our paper.

Point 1: Please add a block diagram of the proposed research step by step, what is the result of the paper?

Response 1: As suggested by the reviewer, a block diagram is added in Section 5.3 representing our proposed system approach. Moreover, we’ve added the explanation of the proposed research step by step at line number 392-402

The experimental analysis, evaluated results are discussed in the section below. Section 6 describes the results achieved by the proposed system.”

Point 2: The technique used in this paper is collaborative filtering which is based on ALS. If the data volume is very large, the transformation will become a super-slow process. Please give out the details about the transformation and address how to handle a large amount of data in this step.

Response 2: Yes able reviewer regarding collaborative filtering is based on ALS. Yes if the data volume is very large the transformation will become a very slow process. For this, We have added these details in the revised manuscript at line number 246-254

our proposed RS is based on user-level and item-level filtering, which is derived from collaborative filtering through a big data framework spark which stores data in-memory in different clusters. Although the dataset is large, we’ve used ALS that decomposed the rating dataset into a matrix, which was item- and user-based along with latent features. Thus, when we recombined them, we got a new matrix by multiplying those decomposed matrices. The new matrix may have a missing entry that represented the movies and predictions as well as may cause a cold-start problem. If there’s any missing entry, the system will fill the space based on user and item level similarities. To avoid the creation of a new model at every instance, we utilized elements that were extracted from the new user-matrix which is attained by simply multiplying the matrices.”

Point 3: Section 1 introduction part is the most important part of the manuscript and authors should need to update the structure of that part that will be good for readers such as AI>>ML>proposed approach with artificial intelligence

Response 3: As pointed by a worthy reviewer, we have revised the complete introduction section from line 39 to 144 to follow the suggested structure.

Point 4: References should be 2018-2021 Web of Science about 50% or more

Khadse, Vivek P., Syed Muzamil Basha, N. Iyengar, and D. R. Caytiles. "Recommendation Engine for Predicting Best Rated Movies." International Journal of Advanced Science and Technology 110 (2018): 65-76.

Khan, M.A., Karim, M. and Kim, Y., 2018. A two-stage big data analytics framework with real world applications using spark machine learning and long Short-term memory network. Symmetry, 10(10), p.485.

Khalid, Z. M., & Zebaree, S. R. (2021). Big Data Analysis for Data Visualization: A Review. International Journal of Science and Business, 5(2), 64-75.

Choudhury, S. S., Mohanty, S. N., & Jagadev, A. K. (2021). Multimodal trust based recommender system with machine learning approaches for movie recommendation. International Journal of Information Technology, 1-8.

Response 4:  These references are updated and new references are cited in the revised manuscript in the references section as well.

Khadse, Vivek P., Syed Muzamil Basha, N. Iyengar, and D. R. Caytiles. "Recommendation Engine for Predicting Best Rated Movies." International Journal of Advanced Science and Technology 110 (2018): 65-76.

citied with [7]  line number 73 and in references section line 572

Khalid, Z. M., & Zebaree, S. R. (2021). Big Data Analysis for Data Visualization: A Review. International Journal of Science and Business, 5(2), 64-75.

citied with [9]  line number 82 and in references section line 576

Choudhury, S. S., Mohanty, S. N., & Jagadev, A. K. (2021). Multimodal trust-based recommender system with machine learning approaches for movie recommendation. International Journal of Information Technology, 1-8.

citied with [30]  line number 202 and in references section line 617

Khan, M.A., Karim, M. and Kim, Y., 2018. A two-stage big data analytics framework with real world applications using spark machine learning and long Short-term memory network. Symmetry, 10(10), p.485.

citied with [47]  line number 537 and in references section line 652

Point 5: The major contribution of the paper looks very weak so authors should think about it deeply as given Line98~101

Response: As suggested by the reviewer, the major contributions with aim have been rewritten in lines 118-135.

The aim is to direct users to items that are most relevant to them. This could be based on past behavior on that platform, or it could be based on the users who have similar profiles to this given user, so the system would recommend something which their friends like which is the field of information extraction. In this regard, we have proposed a movie recommendation system using the big data approach. The main contribution of this paper is as follows:

  • We propose a recommender engine that predicts top ratings for movies. Our system is implemented using collaborative filtering through Apache Spark ML libraries in cloud-based platform data bricks.
  • For this, we applied a model-based approach along with Matrix Factorization and the ALS method, which is used to better learn all latent features. It analyzed the ALS algorithm's parameters which include rank, lambda, and iterations.
  • We achieved an RMSE of 0.9761 with slightly shorter predictions than the existing methods since they only represented the top 25 predictions.
  • We further improved the root mean square error (RMSE) score, hence getting more predictions (going up to 1000) with 97% accuracy, where our algorithm took just 517 seconds to predict these ratings with significant accuracy.

Point 6: How do you measure the success of a recommendation? here is the data complexity issue, the authors did not discuss complex issues in the manuscript.

Response 6: We thank the reviewer for highlighting this important aspect, we have addressed the issue of cold-start and sparsity and added this paragraph (line 102-114) into the introduction section and revised the whole introduction section in which these issues are addressed properly and suggested strategies of our proposed system which resolves these issues through our utilized methodologies.

Point 7: Before Conclusion, please draw a Table and compare with previous researchers, how your approach is better in terms of accuracy. [done]

Response 7: These comparison results are now presented in Table 12. We have added these details to describe the results in terms of accuracy

“our proposed system obtained 97% accuracy with top 1000 predictions with an optimized RMSE score of 0.89-0.97. Besides, existing studies had an RMSE of up to 0.91, but they didn’t mention any accuracy of each iteration or system. We performed two experiments, in each we got predictions with better accuracy and RMSE score (as seen in Table 12).”

Point 8: Authors miss experiment setup, without experimental setup results are doubtful please explain the experimental environment in detail. few parameters are missing?? [done]

Response 8: As pointed by a worthy reviewer, we’ve described the experimental environment in section 6 by adding these lines “and to evaluate all the experimental analysis and results, the environment used is data bricks a cloud-based platform”. We’ve described the experimental environment in section 6.2 in the implementation of the system section about data bricks. Moreover, we have improved section 6 “The experimental evaluation”  by adding regularization and latent factor values in the text paragraphs and results are added at the end of the experiment evaluation and by deleting unnecessary Tables from 06-10 and combined the Table 06-09 in one as Tab. 06.

Point 9: Conclusion: point out what are you done. [done]

Response 9: As suggested by the reviewer we modified the conclusion according to our work done

“In this paper, we applied the ALS (Alternating Least Squared) algorithm problem with CF that solves the scalability, sparsity, and cold-start problem. We compared the results of our developed model with the state-of-the-art by performing two experiments with different divisions of modules in the dataset. In our first experiment, we achieved an RMSE of 0.9761. The 2nd iteration obtained predictions for up to 1000 movies, and a minimum root means squared error (oRMSE) of 0.8951, with improved accuracy of up to 97%. Based on our results, we conclude that the selection of parameters related to alternating least squares could affect the performance of RS.”

Point 10: is there a possibility to use the proposed method for other problems?

Response 10: The proposed model can be used for recommendation systems such as books and news. In the future, the proposed method can be used to achieve significant results particularly in recommendation systems based on the developed approach. Line 537- 538

Reviewer 2 Report

The aytghors provide a study about the collaborative filtering method based on ALS for predicting best-rated movies on the basis of Big Data Architecture.

The proposed study is interesting but there are some points that the authors should investigate.

The Abstract looks more like an introduction. It does not provide a brief of the methodological details neither the results of the experiments conducted for the purposes of this study. The author(s) should revisit. 

The authors should be better described the novelties of their analysis with respect to existing ones The Introduction section should be further revisited by underlining the novelties of the proposed analysis. Figure 1 in the methodology section should be better discussed for underlying the main module for recommendation aim.  In the evaluation section more details about Table 5,6,7,8,9 and 10 should be provided.  Furthermore, the authors should provide more details and discussion about the obtained results. 

I suggest to further analyze more recent approaches about the examined topics. In particular, I suggest the following papers to investigate relevance of multimedia content and Big Data Architecture for recommendation task in the Introduction section:

1) Multimedia recommendation using Word2Vec-based social relationship mining. Multimedia Tools and Applications, 1-17.
2) Kira: a system for knowledge-based access to multimedia art collections. In 2017 IEEE 11th international conference on semantic computing (ICSC) (pp. 338-343). IEEE. 

Finally, I suggest a linguistic revision.

Author Response

Reviewer 2:

Comment

The aytghors provide a study about the collaborative filtering method based on ALS for predicting best-rated movies on the basis of Big Data Architecture.

The proposed study is interesting but there are some points that the authors should investigate:

Response: We would like to thank the reviewer for their valuable time and encouraging comments. The comments have been incorporated as indicated below. Furthermore, we improved the introduction section, language, styles, and figures to make the paper useful for readers. Once again, we do appreciate all your positive comments and valuable suggestions; they helped improve the quality of our paper.

Comment 1: The Abstract looks more like an introduction. It does not provide a brief of the methodological details neither the results of the experiments conducted for the purpose of this study. The author(s) should revisit. [done]

Response 1: As suggested by the able reviewer, we removed these lines

“Machine learning (ML) has become an effective tool with the recent growth of diverse techniques that have been proven helpful for various applications. Commonly there are three filtering methods: content-based, collaborative filtering (CF), and demographic filtering.”

and re-written the abstract more briefly by adding methodological details and experimental results into it and it has been changed to this now:

“In this era of big data, the amount of video content has dramatically increased with the exponential broadening of streaming services, and it has become very strenuous for end-users to get the desired videos. Therefore, to attain an accurate and robust clustering of information, a hybrid algorithm is used to introduce the recommender engine with collaborative filtering using apache spark and ML libraries. In this paper, we implemented a movie recommendation system based on a collaborative filtering method using the ALS (Alternating Least Squared) model to predict the best-rated movies. The proposed system uses the last search data of the user about movie category and reference to instruct the engine and make a list of predictions of top ratings. This study used a model-based approach of matrix factorization, ALS algorithm along with CF technique which solves the cold start, sparse, and scalability problem. This study performed experimental analysis and successfully obtained the minimum root mean squared error (oRMSE) from 0.8959 to 0.97613 approximately. Moreover, the proposed movie recommender system achieved an accuracy of 97% and predicted the top 1000 ratings of movies.”

Comment 2: The authors should be better described the novelties of their analysis with respect to existing ones The Introduction section should be further revisited by underlining the novelties of the proposed analysis. [done]

Response 3: As suggested by the able reviewer regarding novelty by correcting and re-writing the introduction section more precisely by adding research specifications from lines 74-79, line number 81-85 describes how this system providing a better user experience. We’ve added in lines 117-124, a paragraph that discussed the novelty and aim of our research how it produced a robust RS. Then, we’ve added major contributions of the proposed system at lines 125-137, we’ve added experimental analysis observation.

Comment 4:  Figure 1 in the methodology section should be better discussed for underlying the main module for recommendation aim. [done]

Response 4: As pointed out by the reviewer about figure 1 as main module for recommendation aim but at our work figure 05 is regarding the aim of recommendation. Moreover figure 7 is block diagram of our propose work.

“Figure 5 depicts the proposed RS which is based on the user-level and item-level techniques, utilized with matrix factorization approach along with ALS that decomposed the rating dataset into a matrix that was item-based and user-based along with Latent features. Thus, when we recombined them, we got a new user matrix. The new matrix may have a missing entry. If there’s any missing entry, the system will fill the space based on user and item level similarities. It will re-fill the values based on U-matrix.”

Comment: In the evaluation section more details about Table 5,6,7,8,9 and 10 should be provided. [done]

Response 4: As suggested by the able reviewer, the description of tables is added: “It shows iterations with various Latent factors values 06, 08, 10, and 12 respectively. It also describes regularization values and validation Root mean Square Estimation  VRMSE score regarding each factor value.”

We’ve modified this section more and added the regularization values as: 0.5,0.1, 0.2, 0.1, 0.4 and validation the VRMSE score regarding each factor value that can be seen in the Tab. 06. We defined Table number 06 by combining the Table 06-09 and added all the values in the one new Table i.e. Table. 06 (line number 489. We’ve added more detailed about value and added more paragraphs regarding Tables as well as at the end of the experiment.

Whereas the table 10 is changed into table 08 regarding results of our expereimets  (line#  514)    

Comment: Furthermore, the authors should provide more details and discussion about the obtained results. [done]

Response: As suggested by the worthy reviewer, we’ve corrected and mentioned the results in the following form of paragraph in the evaluation section as

The best model has 6 latent factors and regularization=0.2: training RMSE is 0.6876113839813485; validation RMSE is 0.89515 in very less runtime: 517.30 seconds, and obtained the testing RMSE of 0.8959197529794084., we utilized the model and monitored the Precision, training, validation testing scores which predicted the ratings of users’ and obtained the best results for the different number of parameters and iterations.”

The more discussion has been added Line 467 to 500 and Line 516 to 529  respectively.

Moreover, we also created a comparison table of our outperforming accuracy of the system with the exiting accuracy in Table 08 but depicted only top 35 predictions in the Table 07.

RMSE

Accuracy

Prediction

oRMSE1 in exp1

0.9761

93.5%

5

oRMSEin exp2

0.8959

97.0%

1000

Existing RMSE

0.9100

91.0%

25

Comment: I suggest to further analyze more recent approaches about the examined topics. In particular, I suggest the following papers to investigate relevance of multimedia content and Big Data Architecture for recommendation task in the Introduction section:

1) Kira: a system for knowledge-based access to multimedia art collections. In 2017 IEEE 11th international conference on semantic computing (ICSC) (pp. 338-343). IEEE.

2) Multimedia summarization using social media content. Multimedia Tools and Applications, 77(14), 17803-17827.

3) Multimedia story creation on social networks. Future Generation Computer Systems, 86, 412-420.

Response:  As suggested by the able reviewer, we are cited mentioned paper references in the introduction section in the revised manuscript.

The citation [3], [5], [6] in the references section line number 569, 572 and 574

Comment: Finally, I suggest a linguistic revision.

Response: We have significantly improved the grammar, spelling, and sentence structure of the manuscript. We have utilized the MDPI English editing service.

Round 2

Reviewer 1 Report

The authors did excellent work and resolve my previous comments very well now this paper looks very good for readers so I agree to accept this paper for publication.

Reviewer 2 Report

I think that the authors have addressed all my concerns.